# Unsafe Bicycling Behavior in Changsha, China: A Video-Based Observational Study

**DOI:** 10.3390/ijerph17093256

**Published:** 2020-05-07

**Authors:** Yuyan Gao, David C. Schwebel, Lingling Zhang, Wangxin Xiao, Guoqing Hu

**Affiliations:** 1Dongcheng District Center for Disease Control and Prevention, Beijing 100009, China; gaoyuyangyy@csu.edu.cn; 2Department of Epidemiology and Health Statistics; Hunan Provincial Key Laboratory of Clinical Epidemiology, Xiangya School of Public Health, Central South University, Changsha 410078, China; xiaowangxin@csu.edu.cn; 3Department of Psychology, University of Alabama at Birmingham, Birmingham, AL 35294, USA; schwebel@uab.edu; 4Department of Nursing, College of Nursing and Health Sciences, University of Massachusetts Boston, Boston, MA 021125, USA; Lingling.Zhang@umb.edu

**Keywords:** unsafe cycling behaviors, shared bicycle, personal bicycle, China, urban

## Abstract

The recent emergence of shared bikes has inspired renewed use of bicycles in urban China. However, incidence rates of unsafe cycling behaviors have not been reported using objective methods. We designed a video-based observational study in Changsha, China to estimate the incidence of five unsafe bicycling behaviors among both shared and personal bike riders and examine incidence differences across types of riders and cycling areas. A total of 112 h of video recorded 13,407 cyclists riding shared bikes and 2061 riding personal bikes. The incidences of not wearing a helmet, violating traffic lights, riding in the opposite direction of traffic, not holding the handlebar with both hands, and riding in a non-bicycle lane were 99.28%, 19.57%, 13.73%, 2.57%, and 64.06%, respectively. The incidence rate of all five kinds of behaviors differed significantly across four types of riding areas (shopping, university, office, and leisure) and the rates of the first three kinds of behaviors were statistically different between shared and personal bike riders. In situations where bicycle lanes were available, we observed the incidence of riding on the motorway and on the sidewalk to be 44.06% and 19.99%, respectively. We conclude that unsafe cycling behaviors occur with alarming frequency and differ somewhat between riders of shared versus personal bikes. Further research is recommended to interpret the occurrence of risky cycling and the incidence differences across types of riders and cycling areas.

## 1. Introduction

Bicycling was a dominant mode of transportation in urban China from the 1970s through the 1990s [1], but lost popularity in the early 2000s with the growth of automobile traffic. With the broad nationwide introduction of an internet-based bicycle sharing system in 2016 [2], bicycling has suddenly re-emerged as a popular mode of transportation in urban China. By July 2017, there were 16 million shared bicycles placed in Chinese cities and 106 million registered users of those bicycles [2]. Consequently, the proportion of Chinese citizens who reported riding a bicycle more than doubled between 2016 and 2017, and the total distance cycled by shared bicycle riders reached 2.5 billion kilometers in April 2017 [3].

Bicycling offers many societal advantages. From a health perspective, it increases physical activity and prevents disease. It also reduces motor vehicle traffic, alleviating air pollution and traffic congestion and aligning with goals of building an eco-friendly society [4]. At present, the Chinese transportation department encourages and supports the development of bicycle sharing systems [5]. Transportation experts predict bicycle-riding, both using shared and personal bikes, will continue to serve as a central transportation option in urban China for many years [6].

The increased use of bicycles in China is not without negative consequences, however. In the context of continuously increasing motor vehicle traffic, growing use of bicycles increases the risk of road traffic crashes involving bicyclists. A recent survey found that 202 of 2883 (7%) shared bicyclists self-reported 292 road traffic injuries over the past 12 months in Hangzhou, China [2]. Data from the Global Burden of Diseases study estimate deaths from fatal bicycling injuries have increased 104% from 1990 to 2017 (10,969 to 22,376) in China [7].

To develop data-driven interventions to reduce bicycling injury risk in urban China, research is needed to explore behavioral factors that influence cyclist safety. Such research is limited in the current literature. One available study used an internet-based survey among 1960 shared bicycle riders to evaluate the incidence of eight kinds of behaviors in the past month and found the proportion of participants who reported always or often having unsafe riding behavior in the past month ranged from 1.1% for carrying passengers on bicycles to 97.6% for not wearing a helmet [8]. However, online surveys are biased by non-random selection of study participants and poor self-report recall [8]. Other observational research focused primarily on electric bicycle riders in China. One study in Beijing found that 56% of bicycle riders (both electric and traditional bicycles) traveled through red-light signals [9]. Another investigation considered several unsafe behaviors among electric bicycle riders in Suzhou and reported the concerning rates of riding in a motor vehicle lane (1.9%), running red lights (4.8%), riding in the opposite direction of traffic (3.4%), mobile phone use (0.4%), and not wearing a helmet (91%) [10]. Behavior of electric bicycle riders is likely to be different from behavior of traditional bicyclists, however, given their capacity for higher speeds and acceleration [9]. In addition, several studies report behaviors among both shared and personal bicycle riders in other countries [11,12,13,14,15,16]. Robust estimates based on objective observation of unsafe cycling incidence among shared and personal bicycle riders in urban China are absent in the published literature.

We therefore conducted a video-based observational study in Changsha, a large urban Chinese city, to evaluate the incidence of unsafe cycling behaviors and to compare the difference in incidence of unsafe behaviors between shared and personal bike riders and across four kinds of cycling areas (shopping, university, office, and scenic).

## 2. Materials and Methods

### 2.1. Study Location and Participants

This observational study was conducted in Changsha, a large urban city in Southern China. All cyclists appearing at 18 selected sites were recorded by video during peak cycling hours. Observations included both shared bike and personal bike riders.

### 2.2. Ethical Approval

We recorded cycling behaviors—including risky and unsafe behaviors—by video. Private information about cyclists, including their appearance and identities, was carefully protected. The research protocol was approved by the Medical Ethics Committee of Central South University (No. XYGW-2018-40).

### 2.3. Video-Based Data Collection

We collected data in four types of areas where cyclists commonly travel, as identified by a previous report [17]: shopping area (near large shopping malls and stores, accounting for 25% of cycling volume), university area (in and near major universities, accounting for 21% of cycling volume), office area (near large office towers and businesses, accounting for 19% of cycling volume), and scenic area (near scenic spots, accounting for 13% of cycling volume). Residential areas explained the remaining 22% of cycling volume in the previous report [17], but were omitted from this study because they are often adjacent to or intermingled with the other types of areas in Chinese cities.

A total of 18 observation sites were selected: one road segment in the shopping area, two traffic intersections and five road segments in the university area, two traffic intersections and three road segments in the office area, and two traffic intersections and three road segments in the scenic area. The observation sites were selected based on the following criteria: (a) representative of an area; (b) had sufficient space to place a smartphone-based camera and tripod on one side of the street; (c) had no obstacles that obstructed the video of passing traffic; and (d) did not affect typical traffic patterns in any way.

Eleven trained data collectors completed filming in October 2018. Pilot observations were conducted from 7:30 to 20:00 in the four areas to help us select the following peak cycling hours in all areas: 7:30–9:30, 13:30–15:30 and 16:00–18:00 weekdays in the university area; 7:30–9:30 and 16:30–18:30 weekdays in the office area; 8:30–10:30 and 16:00–18:00 weekends in the scenic area. Cycling behaviors were recorded during all peak hours for a single day at each site in the university, office, and scenic areas. In the single shopping area site, pilot observations were made for a full day, but the planned video-based observation was canceled because the police began enforcing traffic laws stringently in that area. Thus, only the one-day pilot video was available for the shopping area.

### 2.4. Video Coding

We treated a one-hour video as a “video hour”. Data collectors listed the date, time, type of observational area and observational point for each video hour. In total, 112 video hours were available for analysis. Based on previous studies [8,15,18], Chinese road traffic safety law [19], and group discussions, we selected five unsafe cycling behaviors for analysis: (a) not wearing a helmet, (b) cycling through a red light, (c) riding against traffic flow, (d) not holding the handlebar with both hands, and (e) riding in non-bike lane. Each of the five behaviors was identified and coded by a trained researcher for each cyclist via manually watching each video hour. All cyclists appearing on the videos were coded; at times, many cyclists were present simultaneously and the coding was completed by rewinding the tape to capture data from all cyclists. Because shared bicycles from each shared bike company in China have unique and distinguishing colors and appearances compared to personal bikes, researchers readily distinguished shared bike riders from personal bike riders in the videos.

Thirty video hours (27%) were selected randomly to be re-coded by another researcher for re-evaluation to establish inter-rater reliability. The consistency coefficient between the two independent researchers was 98.6%, indicating strong inter-rater reliability.

### 2.5. Statistical Analysis

We calculated the incidence and 95% confidence interval of each cycling behavior. Chi-square tests evaluated the differences in incidence of unsafe behaviors between shared and personal bicycle riders and across the four types of study areas. Odds ratios (*OR*) were estimated by fitting logistic regression models. All statistical analyses were conducted in SPSS 20.0. *p* < 0.05 was considered statistically significant.

## 3. Results

The 112 h of video captured 15,468 cyclists at the 18 observation points, including 13,407 shared bike riders (87%) and 2061 personal bike riders (13%) (Table 1). Overall, 5308 cyclists (34%) were recorded crossing traffic intersections and 10,160 cyclists (66%) riding along road segments. Of the 15,468 cyclists, 10,923 (71%) were recorded on roads that had a bike lane available.

15,356 of the 15,468 cyclists recorded (99.28%, 95% *CI*: 99.14–99.41) were observed to be cycling without a helmet. Shared bike riders had a higher incidence of not wearing a helmet than personal bike riders (99.74% vs. 96.26%, χ^2^ = 300.09, *p* < 0.001) (Figure 1). Cyclists in university (99.69%) and office (99.06%) areas had the highest incidence of not wearing a helmet, followed by those in scenic (98.19%) and shopping (97.83%) areas (χ^2^ = 73.20, *p* < 0.001) (Figure 2). Multivariate analysis showed shared bike riders and cyclists in university and office areas had higher risk of not wearing helmet than personal bike riders (*OR* = 18.90, 95% *CI*: 12.59–28.57) and cyclists in scenic areas (*OR* = 9.43, 95% *CI*: 5.42–16.39 and *OR* = 2.27, 95% *CI*: 1.35–3.79, respectively) (Table 2).

1039 of the cyclists we recorded at intersections (19.57%, 95% *CI*: 18.50%–20.64%) cycling through a red light. Personal bike riders had a much higher incidence of cycling through red lights than shared bike riders (46.27% vs. 14.99%, χ^2^ = 412.76, *p* < 0.001) (Figure 1). Additionally, bicyclists in scenic areas had the highest incidence (21.17%), followed by those in university (20.77%) and office areas (0.73%) (χ^2^ = 334.64, *p* < 0.001) (Figure 2). Recordings in the shopping area did not include any intersections. A multivariate model found that shared bike riders (*OR* = 0.63, 95% *CI*: 0.52–0.77) and cyclists in university (*OR* = 0.70, 95% *CI*: 0.56–0.86) and office (*OR* = 0.02, 95% *CI*: 0.01–0.04) areas had a lower incidence of cycling through red lights than personal bike riders and cyclists in scenic areas (Table 2). 

A total of 2124 cyclists (13.73%, 95% *CI*: 13.19–14.27%) were observed to be riding against the traffic flow. Shared bike riders were more likely to do this than personal bike riders (14.72% vs. 7.28%, χ^2^ = 83.60, *p* < 0.001) (Figure 1), and riding against the traffic flow was most common in shopping (24.54%) areas, followed by office (18.27%), scenic (14.31%), and university (10.47%) (χ^2^ = 240.26, *p* < 0.001) areas (Figure 2). A multivariate model found that shared bike riders were more likely to ride contrary to traffic flow than personal bike riders (*OR* = 2.08, 95% *CI*: 1.75–2.48); further, cyclists in shopping and office areas had higher incidence of this behavior than cyclists in scenic areas (*OR* = 1.96, 95% *CI*: 1.59–2.43 and *OR* = 1.33, 95% *CI*: 1.13–1.58, respectively), and cyclists in university areas had lower incidence than cyclists in scenic areas (*OR* = 0.72, 95% *CI*: 0.61–0.84) (Table 2).

We observed 397 cyclists (2.57%, 95% *CI*: 2.32–2.82%) cycling without holding the handlebar with both hands. Personal and shared bicycle riders had similar incidence rates for this risky behavior (2.96% vs. 2.51%, χ^2^ = 1.47, *p* > 0.05) (Figure 1). Cyclists in scenic areas had the highest incidence (3.56%), and those in office areas the lowest (2.02%) (χ^2^ = 11.66, *p* < 0.001) (Figure 2). A multivariate model showed that cyclists in office areas were less likely to cycle with one or no hands on the handlebars compared to those in scenic areas (*OR* = 0.56, 95% *CI*: 0.39–0.80). (Table 2).

For the observation sites where bike lanes were available, 6997 of the 10,923 cyclists (64.06%, 95% *CI*: 63.16–64.96%) were recorded riding in a motor vehicle or pedestrian lane rather than the designated bike lane. Personal bike riders rode in non-bike lanes more often than shared bike riders (67.65% vs. 64.06%, χ^2^ = 90.06, *p* < 0.001) (Figure 1). The incidence of riding in non-bike lanes was highest in university areas (78.27%), followed by scenic (74.00%), office (49.67%), and shopping areas (40.30%) (χ^2^ = 1268.88, *p* < 0.001) (Figure 2). 

We also considered what lanes were selected by cyclists when bicycle lanes were available. Riding on a motorway was the most common selection, accounting for 44.06% of all recorded cyclists in situations where a bicycle lane was available (95% *CI*: 43.13–44.99%). This was followed by riding in a bike lane as required by law (35.94%, 95% *CI*: 35.04–36.84%) and then riding on a sidewalk (19.99%, 95% *CI*: 19.24–20.74%). These selections varied across cycling areas. Riding on the motorway was comparatively common in university and scenic areas, while riding in the designated bike lane was more common in office and shopping areas. A multivariate model showed no statistical differences in incidence between shared versus personal bike riders (*p* = 0.054), but compared to cycling in scenic areas, cycling in the university area had a higher incidence of riding on motorways or sidewalks (*OR* = 1.26, 95% *CI*: 1.06–1.50) while riding in office and shopping areas had lower incidence rates (*OR* = 0.35, 95% *CI*: 0.29–0.41 and *OR* = 0.24, 95% *CI*: 0.19–0.29, respectively) (Table 2).

## 4. Discussion

This study provides updated, reliable incidence estimates of common unsafe bicycle riding behaviors in urban China using objective and rigorous measurement techniques. Unlike previous studies, the study reports incidence differences across type of riders and type of riding areas, offering results with meaningful implications for the development of bicycling injury prevention programs in China and worldwide. Specifically, our video-based observation of cycling behaviors found a widely-varying incidence of dangerous behaviors, ranging from 2.57% of cycling while holding the handlebar without both hands to 99.28% of cycling without wearing a helmet. Incidence rates for risky behaviors differed significantly between shared and personal bike riders and across the four types of areas for cycling we studied.

We observed 99.28% of cyclists not wearing a helmet, a rate comparable to that reported in a WeChat-based self-reported survey (97.6%) in China [8] but far higher than reports from South Korea (80.9%) [20], the United States (14.7–54.5%) [12,15,21,22], Canada (21.9–54.0%) [13,14,16], Italy (78%) [23] and England (44%) [11]. The differences likely reflect a combination of factors in China, including lack of national bicycle helmet laws [24], poor safety awareness among shared bicyclists [25], a perception of bike helmets being uncomfortable to wear [25,26], and lack of provision of helmets for shared bicyclists [14]. Efforts to increase helmet use through legislation requiring mandatory use has been recommended as a highly effective behavior change mechanism, especially since wearing a helmet reduces cycling-related head injuries by 48% and fatal injuries by 34% [24].

We observed an incidence of cycling against red lights (19.57%) that was similar to an observed rate in 2014 for cyclists in the Chinese cities of Nanjing and Kunming (18.7%) [27], but much higher than the 2017 WeChat-based self-reported survey among shared bike riders in Changsha (1.9%) [8] and much lower than observations at intersections in Beijing (50%) in 2012 [9]. The result also differs from the rates in Manhattan, United States (34.3%) [18], Ireland (61.9%) [28], and Italy (62.9%) [29], but is similar to rates reported in Germany (16.3%) [30] and Boston, United States (17%) [15]. Inconsistencies may derive from geographic and cultural differences, different road traffic environments surveyed and observed (e.g., traffic density, road width), and differences in cyclist safety awareness and law enforcement.

Our finding that 64.05% of cyclists traveled in a non-bike lane despite the presence of a bike lane is concerning and much higher than published results from the United States (5–32.6%) [15,18] and self-report survey results from Kunming and Nanjing in China (32.6%) [27]. Efforts to increase the use of bicycle lanes by cyclists have potential to greatly increase cyclist safety. Such changes in China may require cultural shifts and government investment in road infrastructure. Currently, bike lane travel in China is often impeded by parked cars, street vendors, or other obstacles [8]; legislative, policy and enforcement efforts are needed to alter that situation.

We observed an incidence of cycling against the traffic flow of 13.73%, which is substantially higher than previous reports in the United States (7.2% and 7.8%) [18,31] and China (0.2% and 3.4%) [8,27]. Cycling against the traffic flow is particularly dangerous, so our observed rates raise questions about why it was elevated in our sample and how we could change cyclist behavior. The elevations in our sample may be due to a combination of data collection strategies (e.g., observation vs. self-report methods); safety awareness among shared bike riders [11], who comprised 87% of our sample; and poorly-designed road infrastructure which makes it inconvenient to join the traffic flow on a bicycle when turning onto new roadways [31]. Efforts to change behavior may rely on strong enforcement of existing policies in China that cyclists must travel with traffic or on bicycle lanes, plus efforts to raise safety awareness and change normative behavior among the population. Infrastructure changes will be valuable also, including installation of barriers to prevent cyclists from traveling from bicycle lanes to reverse-traffic lanes as well as efforts to remove or reduce obstacles like parked cars and street vendors in the bicycle lanes.

Our data on cyclists traveling with one or no hands on the handlebar are novel to the literature. We detected 2.57% cyclists of riding with such behavior. Informal analysis of the videos suggested there were various reasons cyclists failed to ride with both hands on the handlebars, such as holding a mobile phone or an umbrella, fixing their hair or adjusting their eyeglasses. In other cases, they simply released both hands from the handlebars for no apparent reason. The use of mobile phones while cycling, which has been reported to be 0.7–3% in previous studies [32,33], is particularly concerning. Mobile phone use among cyclists creates not just visual, aural, and cognitive distraction while negotiating traffic [32,33] but the added risk of physical impairment created by cycling with just one (or no) hand(s).

Our comparisons between shared bike riders and personal bike riders yielded different results across the behaviors measured. As in previous studies from other countries [11,12,13,14,15,16], we found cyclists using personal bikes were more likely to wear a helmet than cyclists using shared bikes, although the incidence of helmet use was very low for all cyclists in our study. Higher helmet use among personal bicyclists is logical; they may own helmets and have higher safety awareness [12,14]. Unlike the situation with helmet use where personal bike riders were safer than shared bike users, we found a much higher risk of cycling through a red light among personal bike riders than among shared bike riders. This result contrasts with findings of similar behavior patterns across the two groups in Boston, USA [15], and is difficult to interpret. It may be that personal bike riders are more confident or more familiar with the roadways and traffic patterns. Our remaining assessments yielded no significant differences between personal and shared bike user risk-taking incidence, matching those from the previous study in Boston [15]. It is notable that the proportion of shared bike riders observed was much higher than personal bike riders (87% vs. 13%) in this study, which was very different from studies from other counties [11,12,13,14,15,16]. The difference may be mainly due to the recent introduction of shared bike system in urban China; before that time, the number of bicycles decreased quickly with the rapid development of motorization [4,34]. Due to different cultures for travelling, such an effect is not as significant in other countries as in China.

Our comparison across cycling areas also yielded interesting results with implications for prevention. Cycling against a red light was comparatively low in office areas, for example, likely reflecting the influence of traffic patrols being present during our observations. We also observed a higher incidence of helmet use in scenic and shopping areas, probably because some cyclists in those areas were riding for physical exercise or leisure rather than commuting [8]. Efforts to target helmet use by commuters, perhaps through helmet programs associated with shared bicycles or workplaces, could be effective. Such programs for shared bike riders have been successful elsewhere [35].

The higher incidence of riding on the motorway in university and scenic areas is likely due to a combination of a few factors. First, there was heavy cycling traffic in these areas, which may have caused rushed or quicker riders to seek a faster path along motorways [27]. Second, bike lanes in these areas were sometimes under construction and/or occupied by motor vehicles. Because parking spaces are very limited in China, bike lanes and sidewalks are often occupied by parked cars, especially in university and scenic areas [36].

Last, we found an elevated incidence of riding contrary to traffic patterns in the shopping areas and low incidence of holding handlebars with one or no hand(s) in the office areas. As in the situation with cyclists riding on motorways, the pattern in shopping areas may reflect some cyclists who are running late and therefore take risks by traveling contrary to traffic in order to arrive to their office in time and avoid penalties for late arrival. The pattern in office areas may also reflect the complexity of cycling in those areas, where cyclists faced dense and complex traffic situations.

This study has three major limitations. First, we selected observation sites and times by considering representative sites as well as their feasibility for filming, but the selection was non-random and not fully representative of all cycling locations in the city or country. Although we omitted fully residential areas from our analysis, it is unlikely to significantly change the results since residential areas and the four types of areas are often mixed and comingled in Chinese cities. Second, we relied on video observation techniques that detected many risks, but not all unsafe behaviors. For example, we were unable to accurately detect distracted cycling from wearing earbuds or headphones, and we could not accurately determine whether riding in non-bike lanes was deliberate on the part of the cyclist or was due to the absence of available bike lanes because they were obstructed. Third, owing to the lack of detailed traffic injury data, we did not examine the relationship between unsafe cycling behaviors and cyclist-related crashes or injuries. Future research might work to overcome these limitations and explore the impacts and mechanisms of unsafe cycling behaviors.

## 5. Conclusions

We recorded an incidence of unsafe cycling behaviors that ranged from 99.28% failure to wear a helmet to 2.57% riding the bicycle with one or no hand(s) on the handlebars in Changsha, China. We observed varying rates of behavioral risk between shared and personal bicyclists and across four types of cycling areas. A range of preventive strategies, including legislative, environmental, enforcement, traffic engineering and behavioral measures, should be considered and implemented to improve cycling safety in urban China. Further research is recommended to continue to explore the factors that may lead to high incidence for specific risky behaviors, interpret differences across types of riders and riding areas, and develop innovative, impactful and cost-effective interventions.

## Figures and Tables

**Figure 1 ijerph-17-03256-f001:**
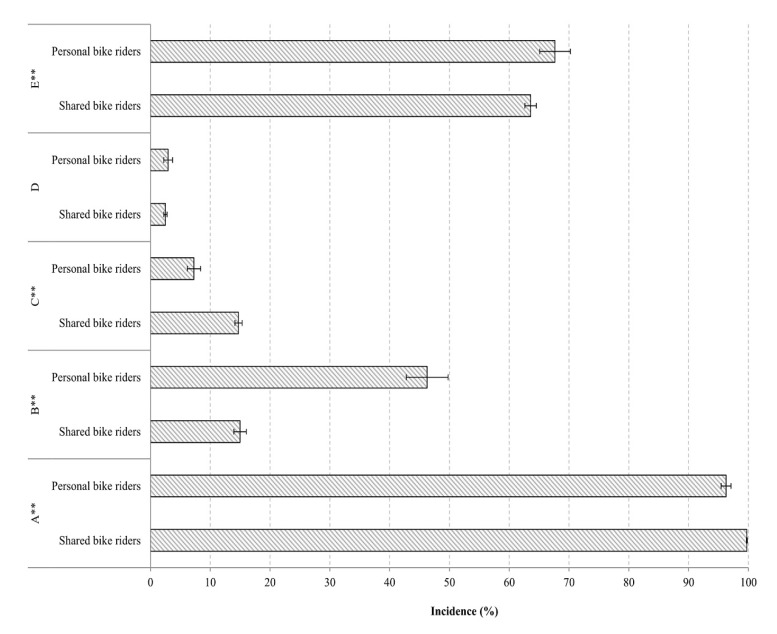
Incidence of unsafe cycling behaviors among shared vs. personal bike riders in Changsha, China. Note: A: not wearing a helmet, B: cycling through a red light, C: riding in opposite direction of traffic, D: not holding the handlebar with both hands, E: riding on motorway or sidewalk. *: *p* < 0.05; **: *p* < 0.01.

**Figure 2 ijerph-17-03256-f002:**
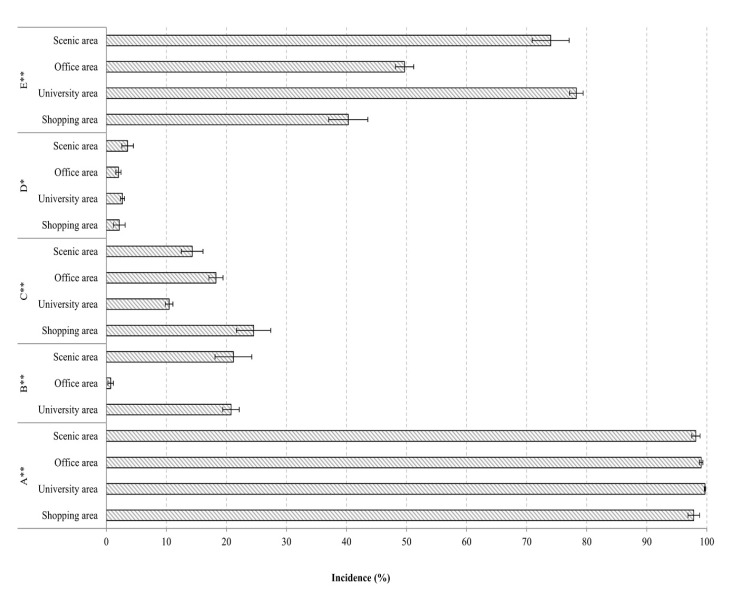
Incidence of unsafe cycling behaviors in four cycling areas in Changsha, China. Note: A: not wearing a helmet, B: cycling through a red light, C: riding in opposite direction of traffic, D: not holding the handlebar with both hands, E: riding on motorway or sidewalk. Behavior B was not observed in the shopping area. *: *p* < 0.05. **: *p* < 0.01.

**Table 1 ijerph-17-03256-t001:** Characteristics of 15,468 observed cyclists in Changsha, China.

Type of Observational Sites	Shared Bike Riders	Personal Bike Riders
Shopping area		
Road segments with bike lane	771	105
University area		
Intersections with bike lane	1529	226
Intersections without bike lane	1259	375
Road segments with bike lane	2916	450
Road segments without bike lane	1900	354
Office area		
Intersections with bike lane	1228	139
Road segments with bike lane	2493	289
Scenic area		
Intersections with bike lane	225	23
Intersections without bike lane	259	15
Road segments with bike lane	448	51
Road segments without bike lane	349	34

Note: Not every type of road segment or intersection was present in every observed area.

**Table 2 ijerph-17-03256-t002:** Associations between unsafe cycling behaviors and both type of cyclist and type of cycling area in Changsha, China (*n* = 15,468).

Variable	Adjusted Odds Ratio (95% Confidence Interval)				
	A	B	C	D	E
Type of cyclist (Ref. = personal bike riders)					
Shared bike riders	18.97 (12.59, 28.57) **	0.63 (0.52, 0.77) **	2.08 (1.75, 2.48) **	0.85 (0.64, 1.12)	0.88 (0.77, 1.00)
Cycling area (Ref. = scenic area)					
University area	9.43 (5.42, 16.39) **	0.70 (0.56, 0.86) *	0.72 (0.61, 0.84) **	0.74 (0.55, 1.01)	1.26 (1.06, 1.50) *
Office area	2.27 (1.35, 3.79) *	0.02 (0.01, 0.04) **	1.33 (1.13, 1.58) *	0.56 (0.39, 0.80) *	0.35 (0.29, 0.41) **
Shopping area	1.03 (0.55, 1.91)	-	1.96 (1.59, 2.43) **	0.6 (0.35, 1.02)	0.24 (0.19, 0.29) **

Note: A: not wearing a helmet, B: cycling through a red light, C: riding in opposite direction of traffic, D: holding the handlebar with one hand or without hand, E: riding on motorway or sidewalk. Behavior B was not observed in the shopping area. *: *p* < 0.05. **: *p* < 0.01.

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
