# Peer review of "Unsafe Bicycling Behavior in Changsha, China: A Video-Based Observational Study"

_ijerph, 2020, doi:10.3390/ijerph17093256_

Round 1

Reviewer 1 Report

This paper reports on a study of unsafe bicycling behaviors in the city of Changsha, China. The data come from 112 hours of video at 18 observation sites in shopping, office, university, and scenic areas. The videos recorded 13,407 cyclists on shared bikes and 2,061 on personal bikes. Five unsafe behaviors were studied and were compared between shared and personal cyclists: failure to wear a helmet, cycling through a red light, cycling against traffic, cycling on the roadway or sidewalk instead of on an available bike lane, and not holding the bike’s handlebar with both hands. Unsafe behaviors for all cyclists ranged from 2.57% for failing to have both hands on the handlebar to 99.28% for cycling without a helmet.

This is a good paper. The subject is important, the data are solid and come from direct observations rather than self-reports or crash reports, the analysis is straightforward, and the results are interesting They add substantially to the available literature on the overall incidence of these unsafe behaviors and on the comparison between cyclists riding shared and personal bikes. The results lead to useful recommendations to improve cyclist safety.

The paper is organized well and written tightly. The writing is generally good, though the paper contains some grammatical errors that can be fixed easily. Some representative examples follow along with a few other suggestions, by line number.

17-19: Are these for all cyclists? Help the reader: perhaps say “overall incidence of …”

18: Grammar and clarity. At 138 you describe the handlebar behavior well but at 18 and 105 you don’t.

21: “Some behaviors …” Which? Again, help the reader.

Further grammar issues at 40 (either “a bicycle sharing system” or “bicycle sharing systems”), 155, 241, 278, 282, and elsewhere. Give the next draft a careful edit.

118-119: This seems to be a high proportion of shared bikes. Do you know how it compares with other areas such as those in the papers you cite in 195ff.

Figures 1 and 2: The labels are hard to read in the small font. Also, use different letters for the two figures if you must use letters, so that A doesn’t mean two things. For that matter, do you need the letters? They are used only in Table 2, where you could use very brief phrases for each column: “no helmet”, “red light”, etc.

209: Isn’t helmet effectiveness proven conclusively rather than just “reported to reduce”?

285 ff: Yes, it would be useful to compare behaviors and crashes. Couldn’t you do this fairly crudely for behaviors that differed substantially across the four areas and that should be coded in crash reports? For example, riding on a motorway or sidewalk?

Author Response

  1. Line 17-19: Are these for all cyclists? Help the reader: perhaps say “overall incidence of …”

Reply: Thanks. The incidences you mentioned involved five types of unsafe behaviors, including not wearing a helmet, violating traffic lights, riding in the opposite direction of traffic, not holding handlebar with both hands, and riding in non-bicycle lane. The incidences of the first four behaviors were for all cyclists, while the incidence of riding in non-bicycle lane was calculated for only the cyclists riding in the situation where bicycle lanes were available. Hence, we think it is better to keep the current terms.

  1. Line 18: Grammar and clarity. At 138 you describe the handlebar behavior well but at 18 and 105 you don’t.

Reply: Thanks. We have improved the description in lines 18, 115 and 159 to make it consistent with the phrase “not holding the handlebar with both hands” in line 153 in the revised manuscript.

  1. Line 21: “Some behaviors …” Which? Again, help the reader.

Reply: Thanks. As suggested, we have specified the unsafe behaviors which significantly between types of cyclist or across four cycling area in lines 20-22 in the revised manuscript. Please see changes below.

The incidence rate of all five kinds of behaviors differed significantly across four types of riding areas (shopping, university, office, and leisure) and the rates of the first three kinds of behaviors were statistically different between shared and personal bike riders.

  1. Further grammar issues at 40 (either “a bicycle sharing system” or “bicycle sharing systems”), 155, 241, 278, 282, and elsewhere. Give the next draft a careful edit.

Reply: Thanks. We have checked and corrected the grammar errors through the whole manuscript. Please see changes in the revised manuscript. As to the use of singular versus plural “a bicycle sharing system” versus “bicycle sharing systems”, there are times when the singular is correct (for example, when discussing the first introduction of a system) and other times when the plural is correct (for example, when discussing multiple systems that are available from multiple companies in the industry).

  1. Line 118-119: This seems to be a high proportion of shared bikes. Do you know how it compares with other areas such as those in the papers you cite in 195ff.

Reply: Thanks. We have added the comparisons for the high proportion of shared bikes with other studies in the discussion and explained possible reasons (lines281-286). Please see changes below.

It is notable that the proportion of shared bike riders observed was much higher than personal bike riders (87% vs. 13%) in this study, which was very different from studies from other counties [11-16]. The difference may be mainly due to the recent introduction of shared bike system in urban China; before that time, the number of bicycles decreased quickly with the rapid development of motorization [4,34]. Due to different cultures for travelling, such an effect is not as significant in other countries as in China.

  1. Figures 1 and 2: The labels are hard to read in the small font. Also, use different letters for the two figures if you must use letters, so that A doesn’t mean two things. For that matter, do you need the letters? They are used only in Table 2, where you could use very brief phrases for each column: “no helmet”, “red light”, etc.

Reply: Thanks. We have increased the size of the figures somewhat to help and assume a copy-editor may work further to increase the size of the writing in the figures prior to publication if this manuscript is accepted to be published. We also agree with the reviewer that it is better to use short phrases to make tables and figures easy to read when possible. However, we are worry the tables and figures would become too large if we do that. We decided to use the letters to represent the five kinds of unsafe riding behaviors and keep this labeling consistent throughout Table 2, Figure 1, and Figure 2. We believe this is preferable to trying to use short phrases but are open to other suggestions from the editor.

  1. 209: Isn’t helmet effectiveness proven conclusively rather than just “reported to reduce”?

Reply: Thanks. We agree helmet use was conclusively proven to be effective for preventing cycling injuries in the systematic review study (reference [24]). We have described it in the revised manuscript (lines 228-231) as suggested. Please see changes below.

Efforts to increase helmet use through legislation requiring mandatory use has been recommended as a highly effective behavior change mechanism, especially since wearing a helmet is reported to reduces cycling-related head injuries by 48% and fatal injuries by 34%[24].”

  1. 285 ff: Yes, it would be useful to compare behaviors and crashes. Couldn’t you do this fairly crudely for behaviors that differed substantially across the four areas and that should be coded in crash reports? For example, riding on a motorway or sidewalk?

Reply: Many thanks. We agree that it would be much valuable to compare riding behaviors and crash. Unfortunately, such data are not publicly available currently in China. We included it as a limitation in the discussion in the revised manuscript (lines 315-316). Please see changes below.

Owing to the lack of detailed traffic injury data, we did not examine the relationship between unsafe cycling behaviors and cyclist-related crashes or injuries.”

Reviewer 2 Report

This study explores the difference in the incidence of unsafe cycling behaviors between shared and personal bike riders and across four kinds of cycling areas. The topic is essential to explore for safe cycling in the context of the rapid development of bike-sharing systems. I hope the following aspects can be taken into consideration to further improve the manuscript.

First, the scientific value of this study is not clearly justified. The authors summarized clearly on different types of unsafe behaviors, but the unique contribution and implications of this study to the existing body of knowledge should be better elaborated.

Second, for the study design, can the authors please more clearly explain the selection of the four sites? As far as I know, shared bikes are largely used for the first and last mile of residents' daily travel, and this includes the travel between residential areas and public transport stations. However, residential areas are not included in the observation sites.

Third, previous observational studies have suggested that video recordings could provide more in-depth observations (Wolfe E S, et al., 2016). The observations in this study are not quite different from non-video-based designs. For instance, one video-based observational study on e-cyclists in a Chinese city also includes gender, traffic condition, mobile phone use, and so on (Yang et al., 2014). Given that this study is entitled "…video-based observational study," I am wondering if more in-depth observations could be provided, or, what are the strengths and novelty that this study could provide compared with previous observational studies.

Fourth, a minor point, the authors explained that riding on a non-bike lane when a bike lane is available might be because the bike lane is occupied by cars and other obstacles (Line 225). In this situation, cyclists have no choice because the bike lane is actually unavailable, and thus this unsafe behavior might be different from other unsafe behaviors (which are more closely related to cyclists' individual safety awareness). Can the authors please further clarify the situations of "bicycle lanes are available"? It's not quite clear about the proportion of cyclists who deliberately riding on non-bike lanes (when the bike lanes are unoccupied) and those who are forced to use non-bike lanes (as the bike lanes are occupied).

Reference:

Wolfe, E.S., Arabian, S.S., Breeze, J.L., Salzler, M.J., 2016. Distracted biking: an observational study. J. Trauma Nurs. 23 (2), 65–70. https://doi.org/10.1097/JTN.0000000000000188.

Yang, J., Hu, Y., Du, W., Powis, B., Ozanne-Smith, J., Liao, Y., Li, N., Wu, M., 2014. Unsafe riding practice among electric bikers in Suzhou, China: an observational study. BMJ Open 4 (1), e003902. https://doi.org/10.1136/bmjopen-2013-003902.

Author Response

First, the scientific value of this study is not clearly justified. The authors summarized clearly on different types of unsafe behaviors, but the unique contribution and implications of this study to the existing body of knowledge should be better elaborated.

Reply: Thanks. As suggested, we highlight the scientific contribution of this study in the first two sentences of the discussion in the revised manuscript (lines 213-217). Please see changes below.

This study provides updated, reliable incidence estimates of common unsafe bicycle riding behaviors in urban China using objective and rigorous measurement techniques. Unlike previous studies, the study reports incidence differences across type of riders and type of riding areas, offering results with meaningful implications for development of bicycling injury prevention programs in China and worldwide.

Second, for the study design, can the authors please more clearly explain the selection of the four sites? As far as I know, shared bikes are largely used for the first and last mile of residents' daily travel, and this includes the travel between residential areas and public transport stations. However, residential areas are not included in the observation sites.

Reply:Thanks for the good suggestion. According to the report in reference 17, 78% of cycling in urban China occurred in the four cycling areas we studied. The remaining 22% occurred in residential areas, which are often adjacent to the four types of areas. In many cases, it is challenging to separate residential areas from the four types of areas, as they are often mixed and comingled in Chinese cities. Therefore, we did not consider residential areas as a category of observation. We have improved our writing to clarify this issue in the revised manuscript (lines 86-93) and also mention the exclusion of residential areas as a study limitation (lines 308-310). Please see changes below.

We collected data in four types of areas where cyclists commonly travel, as identified by a previous report [17]: shopping area (near large shopping malls and stores, accounting for 25% of cycling volume), university area (in and near major universities, accounting for 21% of cycling volume), office area (near large office towers and businesses, accounting for 19% of cycling volume), and scenic area (near scenic spots, accounting for 13% of cycling volume). Residential areas explained the remaining 22% of cycling volume in the previous report [17], but were omitted from this study because they are often adjacent to or intermingled with the other types of areas in Chinese cities.

Although we omitted fully residential areas from our analysis, it is unlikely to significantly change the results since residential areas and the four types of areas are often mixed and comingled in Chinese cities.

Third, previous observational studies have suggested that video recordings could provide more in-depth observations (Wolfe E S, et al., 2016). The observations in this study are not quite different from non-video-based designs. For instance, one video-based observational study on e-cyclists in a Chinese city also includes gender, traffic condition, mobile phone use, and so on (Yang et al., 2014). Given that this study is entitled "…video-based observational study," I am wondering if more in-depth observations could be provided, or, what are the strengths and novelty that this study could provide compared with previous observational studies.

Reply: Thanks for the good point. The primary strength of this study was to provide video-based incidence estimates of unsafe riding behaviors in urban China (especially for shared-bike riders) and examine incidence differences across type of riders and type of riding areas since recent data based on objective measurement are absent for bike riders in urban China. We agree that a rigorous observational design can offer valuable information for in-depth analysis, such as the live observational methods adopted in both Wolfe et al’s study and Yang et al’s study. One primary advantage of our strategy is that we were able to capture behavior of all cyclists through the re-winding and re-viewing of video-recording. At the Chinese sites we studied, there were often many cyclists present simultaneously. Live observers could not possibly gather accurate data from all cyclists and would instead need to rely on some sort of pseudo-random sampling. Because we used videotapes, we were able to capture behavior of all cyclists who were present. We also were able to establish strong inter-rater reliability using the videotapes, something that is more difficult to do with live observations. This increased the rigor of our assessment of the relevant outcomes. Of course, the video-based observational study has its methodological defects also and we now include the limitations in the discussion section.

References:

Wolfe, E.S., Arabian, S.S., Breeze, J.L., Salzler, M.J., 2016. Distracted biking: an observational study. J. Trauma Nurs. 23 (2), 65–70. https://doi.org/10.1097/JTN.0000000000000188.

Yang, J., Hu, Y., Du, W., Powis, B., Ozanne-Smith, J., Liao, Y., Li, N., Wu, M., 2014. Unsafe riding practice among electric bikers in Suzhou, China: an observational study. BMJ Open 4 (1), e003902. https://doi.org/10.1136/bmjopen-2013-003902.

Fourth, a minor point, the authors explained that riding on a non-bike lane when a bike lane is available might be because the bike lane is occupied by cars and other obstacles (Line 225). In this situation, cyclists have no choice because the bike lane is actually unavailable, and thus this unsafe behavior might be different from other unsafe behaviors (which are more closely related to cyclists' individual safety awareness). Can the authors please further clarify the situations of "bicycle lanes are available"? It's not quite clear about the proportion of cyclists who deliberately riding on non-bike lanes (when the bike lanes are unoccupied) and those who are forced to use non-bike lanes (as the bike lanes are occupied).

Reply: Many thanks for the good suggestion. Unfortunately, due to the selection of angle, height and directionality of our cameras, our video data does not allow us to accurately gather information on this issue. Thus, we cannot accurately calculate the proportion of cyclists who deliberately rode on non-bike lane versus those who did so because there were obstacles blocking their pathway in the bike lane. We would note that the safety risks are present either way; the only distinction is the factors that resulted in safety risks. We now include this as a study limitation and thank the reviewer for raising it (lines 313-315). Please see changes below.

and we could not accurately determine whether riding on non-bike lances was deliberate on the part of the cyclist or was due to the absence of available bike lanes because they were obstructed

Reviewer 3 Report

The method of data acquisition and analysis is lacking in the description. Furthermore, it is not clear how the video analysis was done and how the images were decoded to automatically obtain the information.

How the authors recognize the difference between cyclists riding shared bike and cyclists riding personal bikes?

Please, specify how many cyclists have been counted during peak hours for each site.

The authors state that “multivariate model showed no statistical differences in incidence between shared versus personal bike riders”. This conclusion is not adequately supported by the premises and by the description of the data survey methodology: it is not clear how it was possible to distinguish the bikes.

Furthermore, the authors say: “We concluded that unsafe cycling behaviors frequently occurred in riders of shared and personal bikes”. The sentence suggests that there is no difference between the two categories of users. This appears to contradict what is reported in the other sections.

Conclusions and statistics are affected by an inadequate data acquisition methodology.

Author Response

The method of data acquisition and analysis is lacking in the description. Furthermore, it is not clear how the video analysis was done and how the images were decoded to automatically obtain the information.

Reply: Thanks. As suggested, we have included the details about data acquisition and analysis in the revised manuscript (lines 117-119). The images were not really decoded automatically; that work was completed by human coders and we have worked to clarify that in the revision. Please see changes below.

All cyclists appearing on the videos were coded; at times, many cyclists were present simultaneously and the coding was completed by rewinding the tape to capture data from all cyclists.

How the authors recognize the difference between cyclists riding shared bike and cyclists riding personal bikes?

Reply: Thanks for this question, which would be answered easily by people in China but could be confusing to those in other countries. In China, there are several companies offering shared bike services. For marketing purposes, each one uses highly unique and notable appearances for their bicycles (bright colors and notable designs) that are easy to identify. Trained researchers watched the videos to identify the shared bike riders in this study, and given the uniqueness of the shared bicycles this aspect of the coding task was comparatively easy. We have added the process of recognizing shared bike riders in the revised manuscript (lines: 119-122). Please see changes below.

Because shared bicycles from each shared bike company in China have unique and distinguishing colors and appearances compared to personal bikes, researchers readily distinguished shared bike riders from personal bike riders off the videos.

Please, specify how many cyclists have been counted during peak hours for each site.

Reply: Thanks. Data for all cyclists present during the peak hours were coded and counted. No cyclists who were captured on the video were omitted. We now clarify this detail and report the data concerning sample sizes in Table 1.

The authors state that “multivariate model showed no statistical differences in incidence between shared versus personal bike riders”. This conclusion is not adequately supported by the premises and by the description of the data survey methodology: it is not clear how it was possible to distinguish the bikes.

Reply: Thanks. As described in response to the comment above, shared bike riders in China are very easily distinguished from personal bike riders by video. We recognize this would be obvious to most people in China given the bright colors and distinguishing design of shared bicycles created for marketing purposes in China, but would not be obvious to others. To clarify this issue, we have improved the description of methods in the revised manuscript (lines 119-122).

Furthermore, the authors say: “We concluded that unsafe cycling behaviors frequently occurred in riders of shared and personal bikes”. The sentence suggests that there is no difference between the two categories of users. This appears to contradict what is reported in the other sections.

Reply: Thanks. We are sorry for the confusing writing. In fact, the incidence rates of unsafe cycling behaviors were high for both shared and personal bike riders although there were significant incidence differences for specific behaviors between two types of riders. We have improved the wording in the revised manuscript (line 26). Please see changes below.

We concluded that unsafe cycling behaviors frequently occur with alarming frequency and differ somewhat between riders of shared versus personal bikes.

Conclusions and statistics are affected by an inadequate data acquisition methodology.

Reply: Thanks. We respond to the reviewer’s concerns in the above responses and hope this shows the reviewer that our data acquisition methods were scientifically rigorous and reliable.

Round 2

Reviewer 3 Report

Thank you for replying to all comments.